# Refinement of Weak Annotations for the Segmentation of Bone Marrow Leukocytes

Philipp Gräbel[1], Martina Crysandt[2],
Reinhild Herwartz[2], Melanie Baumann[2], Barbara M. Klinkhammer[3],
Peter Boor[3], Tim H. Brümmendorf[2], and Dorit Merhof[1]

[1] Institute of Imaging and Computer Vision, RWTH Aachen University, Germany
{graebel,merhof}@lfb.rwth-aachen.de
[2] Department of Hematology, Oncology, Hemostaseology and Stem Cell
Transplantation, University Hospital RWTH Aachen University, Germany
[3] Institute of Pathology, University Hospital RWTH Aachen University, Germany

**Abstract.** Deep neural networks are well suited to address medical problems such as the automated analysis of leukocytes in bone marrow images. However, their training requires large annotated datasets.
The shortage of annotations is one of the most prevalent problems in biomedical image analysis. Particularly with polygonal contours as segmentation training data, creating high quality annotations is infeasible. Weak annotations, e.g. bounding boxes, can be obtained more easily.
This paper investigates several approaches that aim at refining weak annotations. The resulting refined contours are used to train a semantic segmentation network. Our evaluation shows that it is possible to achieve precision close to training with ground truth data, with a novel U-net method, presented in this paper.

**Keywords:** Leukocyte Segmentation · Weak Annotations · Annotation Refinement · Bone Marrow Microscopy

## 1 Introduction

In deep learning, it is often difficult to acquire a suitable number of annotations, particularly for supervised segmentation of medical images. This makes it necessary to appropriately balance the trade-off between annotation accuracy and the required effort. Therefore, a viable approach is the refinement of quickly obtainable, imprecise annotations prior to network training [8].

An important medical image analysis task is the segmentation of leukocytes (white blood cells). Leukocytes are an integral part of the human immune system. They are created through differentiation of hematopoietic stem cells in the bone marrow. Based on their morphology, leukocytes can be grouped into several different classes in varying stages of maturity. The distribution of cell types is an important diagnostic tool and a deviation from the normal distribution indicates diseases such as leukemia. Conventionally, the required statistics are obtained manually, which is extremely time-consuming and prone to human error. Consequently, an automated analysis tool is desirable.

Even though classical methods exist, they mostly tackle subproblems, such as nuclei segmentation [13], require some form of supervision [15], consider only peripheral blood [16] or a subset of cell types [14]. Deep learning has the potential to overcome these limitations, as has been done in classification tasks [6]. However, there are no public datasets available with a sufficiently large number of cell annotations, particular none which are suitable for segmentation. Creating such a dataset manually is extremely labour intensive. Optimizing this process is a crucial step in building a solid data basis for bone marrow image analysis.

In the category of *Weakly Supervised Learning*, many improvements focus on adapting the network structure or the training procedure. BoxSup [5] iteratively updates an initial bounding box to create more accurate masks during training. Another algorithm using expectation maximization based foreground estimation within a bounding box yields good results [11]. Khoreva et al. suggest that it is also possible to work with a refinement procedure of weak annotations [8].

We propose to use a limited set of expert contoured cells to refine a larger set with weak annotations. This alleviates the need for changing the network structure or training process.

Figure 1 shows the proposed pipeline. In Stage 1 (Annotation Refinement), we investigate several algorithms for the refinement of weak annotations. We employ classical segmentation algorithms, model-based approaches and a novel U-Net based method. In Stage 2 (Network Training), we train a segmentation network on the refined annotations from Stage 1. Theoretically, any neural network could be used in this stage because we use the refined annotations as if they were manual annotations.

**Contributions** (1) We compare classical, model-based and learning-based approaches for the refinement of weak annotations. (2) We introduce a refinement method based on the U-Net segmentation algorithm that yields highly effective training data. (3) We propose a pipeline that exploits a small amount of well annotated data to increase performance on weakly annotated data.

## 2    Image Data

**Dataset** We digitized Pappenheim stained bone marrow samples with $63\times$ magnification using immersion oil. A total of 4500 cells were annotated by medical

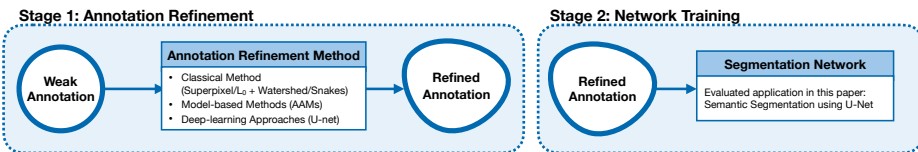

**Fig. 1.** Stage 1 employs one of several different methods for the refinement of a weak annotation. Stage 2 uses the refined annotation to train a neural network.

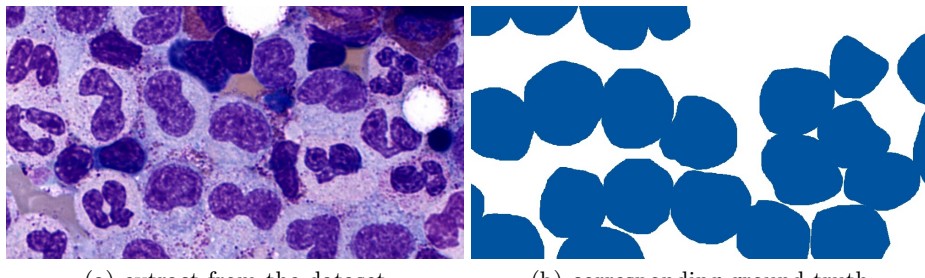

(a) extract from the dataset            (b) corresponding ground truth

**Fig. 2.** An example image with corresponding ground truth from the dataset. Note that the ground truth image is generated from instance polygons.

experts as polygonal contours and class labels. The corresponding segmentation masks denote hematopoietic cells as shown in Figure 2. We extracted 5602 image patches of size $256 \times 256\,\text{px}^2$ from these annotated whole-slide images.

**Weak Annotations** We simulate circular, weak annotations by perturbing the minimal enclosing circle of a cell's ground truth contour. The circle is artificially distorted by adding noise to the center coordinates and the radius to simulate inaccuracies in manually drawing such a circle. The radius is increased to cover the whole cell again, before applying additional noise.

This mimics the characteristics of easily obtainable weak annotations by medical experts: it is easy to define a circle covering the whole cell but difficult to pinpoint the exact center of the minimal enclosing circle.

## 3   Annotation Refinement Methods

The following section captures three different kinds of refinement approaches – classical, model-based and learning-based segmentation. These methods aim at refining an initial, weak estimate of a single cell contour. All refined annotations together form the training set, which is used in Stage 2.

**Classical Segmentation** Each of the following classical pipelines comprises a pre-processing and a segmentation step. For pre-processing, we employ either the commonly used *Superpixel* [1] algorithm or $L_0$-*Smoothing* [17]. Additionally, histogram equalization and low pass filtering are performed. In the segmentation step we use either an *Active Contour Model* [7] (referred to as *ACM* or *Snakes*) or a variant of *Watershed* [3], with the weak annotation as the initial contour. Both are classical methods often used for finding boundary pixels in images. This results in four methods, combining either *Superpixel* or $L_0$-*Smoothing* with either *Watershed* or *Snakes*.

**Model-based Segmentation** A commonly used model-based algorithm for segmentation tasks is *Active Appearance Model Fitting* (AAM) [10]. With *Lucas Kanade*-based minimization [2], landmarks in a given image can be localized by iteratively refining the weak annotation. Ground-truth segmentation polygons are resampled to a fixed number $n_{\text{landmarks}}$ of equidistant landmarks. We trained three different *Patch-based* models: one with $n_{\text{diag}} = 50$ and $n_{\text{landmarks}} = 18$ (denoted as $AAM$), the same based on HoG-Features ($HOG\text{-}AAM$), and a large version with $n_{\text{diag}} = 200$ and $n_{\text{landmarks}} = 54$ ($XL\text{-}AAM$). Furthermore, we evaluate the first AAM with constrained parameters [4] ($C\text{-}AAM$).

**Deep Learning-based Segmentation** In the category of deep learning-based approaches, we chose vanilla U-Nets [12] with a depth between 3 and 6. While this model is often used for segmentation of bio-medical images, it is not directly suitable for *instance segmentation*, as required in this case. Thus, we modify the training data by eroding the ground truth polygon with a circular kernel of fixed size. This ensures that adjacent cells can be distinguished more easily. For refinement, we perform U-Net prediction and dilate the predicted area within the weak annotation using the same kernel as before.

## 4 Evaluation

We perform two experiments to evaluate the precision of the refinement methods and their ability to improve the performance of a segmentation network.

**Stage 1: Annotation Refinement** We compare refined annotations based on the methods presented in Section 3, as well as the weak annotations, with the corresponding manual annotations (ground truth). We employ 3-fold cross-validation, repeated 10 times with different random seeds for the perturbation of the weak annotations.

**Stage 2: Network Training** Further, we evaluate the capability of refined annotations to improve segmentation performance compared to using weak annotations directly. We also test the performance of manual annotations. We chose semantic segmentation using a vanilla U-Net of depth 5 as a network for the second stage.

For this experiment, we separated the dataset into six subsets. In every cross-validation, one of them serves as the test set and the remaining five sets as the training data. For each of the five sets, we generate refined annotations based on training on the other four sets.

**Limited Training Data** We further evaluated the method with limited training data for the first stage. For this scenario, we used one of six subsets as training data for the Stage 1 U-net method (depth 5). Based on this we refine four subsets, which are then utilized as training data for a Stage 2 U-Net (depth 5).

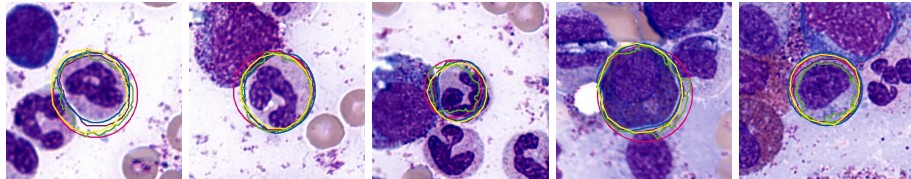

**Fig. 3.** First stage refinement results: $L_0 + WS$ (green), *AAM* (yellow), *U-Net 5* (blue), weak (red), manual annotation (violet)

**Parameters and Setup** The (hyper-)parameters set to the following based on an initial analysis of a wide range of parameters on a smaller subset.

(SLIC) $n_{\text{segments}} = 100$, $c = 10$ ($L_0$-Smoothing) $\lambda = 0.02$ (AAM) 2 scales, 20 shape components, 100 appearance components, $16 \times 16\,\text{px}^2$ patch-size, $10\,\%$ weight for the constraints (U-Net) $8\,\text{px}$ kernel size. Gaussian noise for the weak annotations used $\sigma_{\text{c}} = 7\,\text{px}$ and $\sigma_{\text{r}} = 2\,\text{px}$.

In Stage 2, we only evaluate selected approaches and weak annotations. We train for 150 epochs using an Adam optimizer (learning rate 0.0001) and the Dice loss. We retain the model with the highest validation Dice score. Further, we apply random crop and random rotation for data augmentation and pre-train the U-Net on the medical image dataset of the *MoNuSeg*-Challenge [9].

## 5    Results

We use the Dice score as a criterion to measure segmentation accuracy. Results are shown in Figure 4 a (Stage 1) and Figure 4 b (Stage 2). The refinement results (Stage 1) for five cells are shown in Figure 3.

**Computation Times** Computations on a GTX 1080 Ti and a Quad-core i7 4.6 GHz took $0.1\,\text{s}$ to $0.7\,\text{s}$ for classical methods and normal model-fitting, almost $3\,\text{s}$ for a large or feature-based model and $8\,\text{ms}$ to $17\,\text{ms}$ for the *U-Net* approach.

**Stage 1: Annotation Refinement** Figure 4 a clearly shows that model-based approaches perform better than classical approaches, while deep learning-based methods outperform both. Both classical methods and AAMs generally result in a higher number of accurately fitted cells. The *U-Net* approach yields more precise results, but fails for approximately $10\,\%$ of the cases (not shown in the plot). A failure can occur if no cell is predicted at all or two cells are inseparable. However, they can simply be replaced with weak annotations for the subsequent learning task if this is detected. A deeper network results in a lower failure rate.

Another observation is that the larger AAM (*XL-AAM*) performs worse than the normal-size version (*AAM*). *C-AAM* performs slightly better, while *HOG-AAM* is the most successful model-based approach. Of the classical approaches, $L_0$-preprocessed images tend to yield more accurate segmentations, especially together with Watershed.

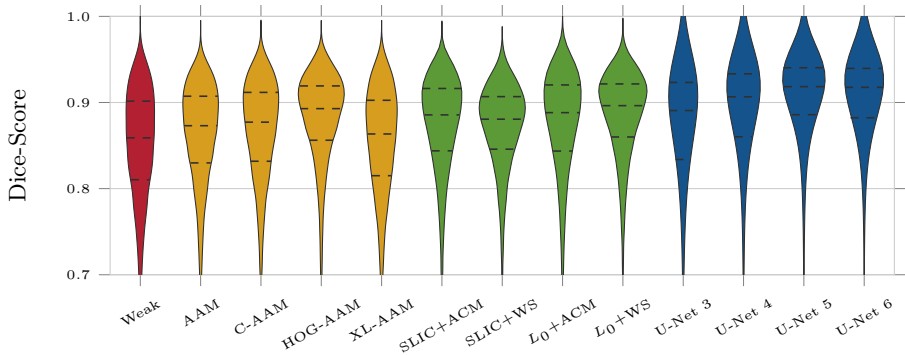

(a) Dice score distribution for the Stage 1 refinement methods.

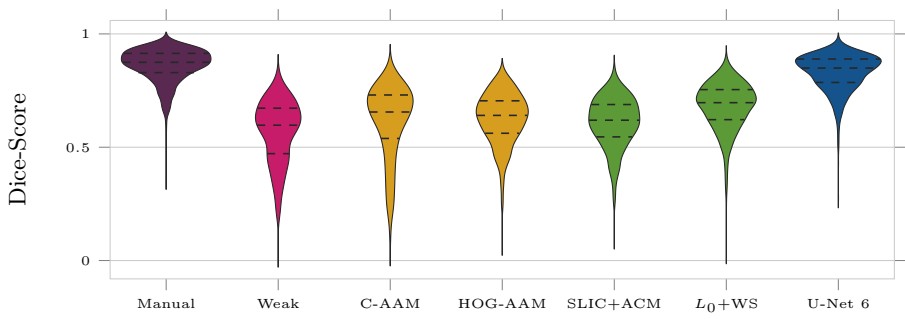

(b) Dice score distribution for Stage 2 segmentation results.

**Fig. 4.** Results for Stage 1 and Stage 2.

**Stage 2: Network Training** Figure 4 b shows the result of the segmentation network trained with different annotation refinement methods.

Even though the weak annotations are quite imprecise, the results are not significantly worse compared to *SLIC+ACM*, which has much higher accuracy in the first stage. The *C-AAM* approach, which is less accurate in segmenting single cells, performs better compared to this classical approach. *HOG-AAM*, which yields better results in the first stage, is not much better than *C-AAM*. $L_0+WS$ is the most accurate approach of the classical and model-based approaches evaluated in this experiment. The *U-Net* method yields results close to using manual annotations.

**Limited Training Data** Using limited training data (i.e., just one of six subsets) for training the first stage, reached similar results as previous experiments: across all folds, we achieved a mean dice score of 0.859 with a variance of 0.0026 in the second stage.

# 6    Discussion & Outlook

The results show that a refinement using classical and model-based approaches yields slightly better Stage 2 segmentation results compared to weak annotations exclusively. The accuracy of the segmentation network is not directly related to the accuracy of the refinement method: some more precise methods yield worse results when used as training data (e.g. *C-AAM* compared to *SLIC+ACM*). Imprecisions in the refined annotations, particularly from model-based and classical approaches, tend to focus on salient image characteristics. Consequently, erroneous annotations created by those approaches are more likely to mislead the segmentation network than random weak annotations.

The success of the *U-Net* approach indicates that by using a method trained on precise contours for refinement of weak annotations, the annotation effort can significantly be reduced without too high loss in segmentation precision. Even though it needs to be ensured that a sufficient amount of training data is available, our results with limited training data indicate that already a small amount is beneficial. We will conduct further research regarding performance as a function of training data. Due to the capabilities of the U-Net in general, it is likely that this approach could be useful to a wider range of applications from different modalities as well as different object shapes. Further improvements could be gained by using a three class network (background, cell, cell contour).

As the process of manually drawing contours is extremely time consuming, it was not feasibly to perform an intra- and inter-rater analysis yet.

We employ the *U-Net* method for the generation of annotated leukocytes in a bone marrow microscopy dataset. This approach significantly reduces annotation time without significant loss in terms of precision in subsequent tasks.

# 7    Conclusion

In this paper, we analyze three different classes of approaches for contour refinement with respect to their suitability to provide data for supervised training of a deep segmentation network. Classical methods and simple model-based approaches improve results when used as training data. Furthermore, we propose a novel method based on U-Net prediction and morphological operations. This method provides refined annotations, which yield segmentation accuracies close to those achieved by networks trained on manually generated ground truth data. We demonstrate the effectiveness of refining weak annotations prior to training on a challenging problem in medical image analysis.

**Acknowledgements** This work was supported by the German Research Foundation (DFG) under grant no. ME3737/3-1. Additional support came through the grants SFB/TRR57, SFB/TRR219, BO3755/6-1, and the RWTH Interdisciplinary Centre for Clinical Research (IZKF: O3-7).

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
