# OpenReview forum: "Refinement of Weak Annotations for the Segmentation of Bone Marrow Leukocytes"
_MICCAI.org/2019/Workshop/COMPAY — COMPAY 2019_

### Official Review · AnonReviewer3 · 2019-07-31
**Mostly an applications paper, but good experimental method on interesting problem. Realistically baselines would be expected to do badly.**

**Rating:** 6
**Confidence:** 4

**Review:**

The premise of this paper is that inaccurate annotations (circles, rather than contours) can be used as training data for cell detection if suitably refined. This is not a very controversial claim, although the particular problem is challenging due to the inhomogeneity of the cells in question. The paper compares several baselines that I wouldn't expect to do particularly well on this complex data (watershed, snakes, AAMs) to a CNN approach. The CNN approach (unsurprisingly) does best. While the novelty in techniques is not high, and the observations all keep with expectations (e.g. random perturbations are  less of an issue than systematic error), this paper is reasonably well written and the experimental methodology well carried out.

Question: Why did you use a 2 class network for segmentation? Several authors have demonstrated a 3 class network (FG, BG, Edge) works well. This could have been achieved with no extra manual effort as a modification to your erosion step. It might be worth discussing this.

Observation: You state that creating contours is infeasible, but then do exactly that (then turning them into circles). Does this mean it is feasible? Or that you have insufficient training data? If the latter how would more data effect the results and conclusions?

---

### Official Review · AnonReviewer2 · 2019-08-11
**Needs a lot more explanation of refinement method**

**Rating:** 5
**Confidence:** 3

**Review:**

Summary:
The authors present a method for segmentation of bone marrow leukocytes, that consists of first refining a set of weak annotations and then utilises these refined annotations for a superior performance.  The authors claim that the weak annotation in the form of a circle can be easily obtained, whereas the minimum enclosing polygon can be difficult to obtain.

Detailed comments and concerns:
- First, I am not fully convinced that the set of ground truth annotations are much more difficult to obtain than the circles. However, this completely depends on the type of tool used for the annotation. For example, a polygon tool would be time consuming, but using a circle tool with a fixed radius would make more sense. This needs clarity. A comparison of the times to obtain the weak annotations and the ground truth annotations would be a good addition.
- Next, I think the refinement step is a little unclear. How are the weak annotations used within the model? Are they used as an extra input or are the weak annotations used as the target. This also needs much more clarity - I think a concept diagram of the refinement step in particular would be useful. This could be combined into Figure 1. I see the refinement stage as the main contribution of the paper and therefore it should be crystal clear exactly what is happening.
- For the segmentation results, the comparative analysis is insufficient. More comparison with other deep learning methods is needed.

Overall, the paper has potential but it needs a lot of work. Stage two of the model is not novel because a simple U-Net is used, but the refinement of weak annotations could be further developed. As mentioned above, focus on the refinement of weak annotations and give more explanation into the method.

---

### Official Review · AnonReviewer1 · 2019-08-14
**No title**

**Rating:** 5
**Confidence:** 3

**Review:**

This work presents a comparison of multiple approaches for refining weak annotations in segmenting leukocytes. The study is conducted on 4500 bone marrow cells where annotations were collected as polygonal contours and class labels. Different types of refinement approaches used in this work are categorized as classical, model-based and learning-based segmentation. The presented results suggest the U-Net has outperformed classical and model-based approaches trained on weak labels and U-Net produces comparable results for models that were trained on manually generated ground truth.

- How the proposed approach is better than Qu, Hui, et al. "Weakly Supervised Deep Nuclei Segmentation using Points Annotation in Histopathology Images." International Conference on Medical Imaging with Deep Learning. 2019.
A comparative analysis with the above approach (their source code is available) would strengthen the paper or at least it should be discussed.
- It was mentioned that a novel U-net method was used but the novelty was not clearly explained in the paper. If authors have made any changes to the existing U-Net model then they should properly highlight those with quantitative results (if applicable).
- Is there any previous work that shows the prognostic or predictive significance of leukocytes segmentation for bone marrow patients?
- In Fig 4 (especially 4b), it is difficult to quantitatively identify the performance difference among different methods so it would be worth embedding the dice score (numeric values) in that figure.
- Just a minor comment, in section 1, paragraph 2, important diagnostic too -> important diagnostic tool

---

### Decision · Program_Chairs · 2019-08-20

Accept